# Hybrid of deep learning and exponential smoothing for enhancing crime forecasting accuracy

Umair Muneer Butt[1,2☯*], Sukumar Letchmunan[1☯*], Fadratul Hafinaz Hassan[1], Tieng Wei Koh[3]

1 School of Computer Sciences, Universiti Sains Malaysia, Penang, Malaysia, 2 Department of Computer Science and Information Technology, University of Chenab, Gujrat, Pakistan, 3 Department of Software Engineering and Information System, Universiti Putra Malaysia, Selangor, Malaysia

☯ These authors contributed equally to this work.
* umair@student.usm.my (UMB); sukumar@usm.my (SL)

**Data Availability Statement:** The New York City crime data used in this study is publicly available here: (https://data.cityofnewyork.us/Public-Safety/NYC-crime/qb7u-rbmr). Moreover, the Python files to implement the proposed methodology are

## Abstract

The continued urbanization poses several challenges for law enforcement agencies to ensure a safe and secure environment. Countries are spending a substantial amount of their budgets to control and prevent crime. However, limited efforts have been made in the crime prediction area due to the deficiency of spatiotemporal crime data. Several machine learning, deep learning, and time series analysis techniques are exploited, but accuracy issues prevail. Thus, this study proposed a Bidirectional Long Short Term Memory (Bi-LSTM) and Exponential Smoothing (ES) hybrid for crime forecasting. The proposed technique is evaluated using New York City crime data from 2010–2017. The proposed approach outperformed as compared to state-of-the-art Seasonal Autoregressive Integrated Moving Averages (SARIMA) with low Mean Absolute Percentage Error (MAPE) (0.3738, 0.3891, 0.3433,0.3964), Root Mean Square Error (RMSE)(13.146, 13.669, 13.104, 13.77), and Mean Absolute Error (MAE) (9.837, 10.896, 10.598, 10.721). Therefore, the proposed technique can help law enforcement agencies to prevent and control crime by forecasting crime patterns.

## 1 Introduction

Providing safety and security to citizens by controlling and preventing crime is crucial for law enforcement agencies. Crime can cause direct or indirect damage to social welfare and can negatively impact the nation's economy [1, 2]. Therefore, countries spend a significant amount of their Gross Domestic Product (GDP) on law enforcement agencies to control and prevent crime [3]. However, little effort has been made in this area due to spatial and temporal information unavailability in crime datasets. But, this area has been getting attention with technological advancement for the last few years. The inclusion of space and time information in the crime datasets using Geographic Information Systems (GIS) enhanced the spirit of the crime prediction research area. Therefore, several efforts have been made in the crime prediction area to assist law enforcement agencies in the last few years [4–6].

available at Github here: (https://github.com/umairmuneer/Crime-forecasting).

**Funding:** This work was supported by the Ministry of Higher Education Malaysia for Fundamental Research Grant Scheme (FRGS) with Project Code: FRGS/1/2020/TK03/USM/02/1, School of Computer Sciences and University Sains Malaysia (USM).

**Competing interests:** The authors have declared that no competing interests exist.

Meanwhile, few European countries, including Germany and Switzerland, use forecasting police programs. Other countries, including Italy, are still evaluating the potential benefits [7, 8]. They use artificial intelligence (AI) models to predict different types of crime in all societies across the country daily, which results in more effective distribution of police forces; this approach focuses on areas of highest risk according to the program [9, 10].

Crimes are typically recorded as a series of observations in chronological order over time. Time series are always individually processed before being combined with time-series parameters in forecasting models. These models can often conduct accurate forecasting but fail occasionally. However, the statistical methods assume linear modelling [11]; thus, they do not extend well and fail to capture patterns. Nevertheless, officials and decision-makers are working on forecasting, limiting, and preventing crime to reduce social harm and secure the state. Furthermore, forecasting and preventing crime is preferable to investigate the course of crime, which has changed in our modern era.

Besides, time series analysis is a popular statistical technique that can forecast the future supply and demand for a service or commodity. The time series analysis method observes a phenomenon (or variable) over a specific period (e.g., several years). It is then predicted based on the different values in the time series and the pattern of value growth. As a result, it outperforms the conventional method [10]. Time-series techniques are used in most fields, including economics, stock and gold price forecasting, energy, pressure, weather forecasting, and crime forecasting [12]. However, given that the goal of time series analysis is to obtain reliable and accurate forecasting, they also require additional information from the series, such as level and seasonality.

Neural networks are also used to solve the problem of time series forecasting. An example is Long Short Term Memory (LSTM), which can preserve the temporal information obtained from the last time but lacks the use of additional time-series parameters such as level and seasonality. Therefore, the forecasting performance with neural networks can be improved by removing seasonality from the primary time series [13]. The study by Nelson et al. [14] using 68 monthly time series from M-Competition showed that de-seasonalization time series data achieve better forecasting performance with neural networks than the seasonal one [15]. Ben Taieb et al. [16] reached a similar conclusion based on an experiment on NN5 data. As a result, achieving large-scale time series forecasting with deep learning is still challenging.

Makridakis, Spiliotis, and Assimakopoulos [17] demonstrated that forecasting in the combined time series is superior to other methods in general. Considerable research is available on time series forecastings, such as autoregressive integrated moving average (ARIMA) for understanding linear patterns and artificial neural network (ANN) for nonlinear patterns [18]. However, the best-performing approach is the hybrid of Exponential Smoothing (ES) and LSTM proposed by S.Smyl [19]. They allow time-series parameters such as seasonality and others to fit the RNN weights. The model does not simply combine classic ES and neural networks in an ensemble manner; instead, it allows all parameters, including ES seasonality and smoothing parameters, to fit concurrently with RNN weights in the same gradient descent method. Thus, it enables accurate time-series specific learning with neural networks. It is a hierarchical model that may be used to forecast numerous series. But, accuracy issues still prevail that need the attention of the researchers.

Furthermore, crime times vary, including hourly crimes such as petit theft and harassment and weekly, seasonal, or annual crimes such as assault, criminal mischief, and related offenses. This paper proposes a hybrid method that combines time series and neural network techniques inspired by the M4 competition [19]. Moreover, this study aims to contribute toward achieving the following goals:

- First, this study proposes a novel hybrid method of Bi-LSTM and Exponential smoothing.

- Second, crime forecasting accuracy is enhanced by fine-tuning ES and Bi-LSTM parameters.

- Third, according to the author's knowledge, forecasting is done according to critical crime types, which is the first attempt.

The remainder of the paper is organized as follows: Section 2 discusses available literature reviews on temporal crime prediction. Next, sections 3 and 4 present the proposed methodology for crime forecasting with algorithmic details and the dataset used for experimental evaluations. Finally, section 5 highlights the significance of the proposed work in a particular experimental setup, and the paper concludes in Section 6.

## 2 Related work

Several efforts have been made in the literature on crime forecasting. This section discusses the most recent and prominent state-of-the-art research. Mainly, time series analysis and prediction techniques are discussed. Moreover, efforts made in regression, deep learning, and hybrid models are highlighted with their advantages and disadvantages.

### 2.1 Time series analysis for crime forecasting

Time series analysis extract time-series information related to crime and crime trends. It can also determine the growth of the existing changes observed simultaneously. Furthermore, it can use analysis methods to determine whether time series data are constant or seasonal. Wawrzyniak et al. [20] proposed that crime has seasonal ups and downs and can be determined. Moreover, the summer, winter, and holiday seasons can affect the model's predictive accuracy. Jha et al. [8] discussed that the time-series approach is a dependable method for understanding data over a long period and at different times (hourly, weekly, quarterly, or yearly). The measurement trend becomes smoother when more observations are used, which results in more accurate forecasting. An essential advantage of the time series approach is detecting seasonal patterns, which is critical for future forecasting.

Linning et al. [21] examined the variation in crime across the year to determine a seasonal trend. They conducted their research using crime data from three Canadian cities to emphasize property-related crimes. The results are expected to show a quadratic relationship because their analysis focuses on crime seasonality. For example, crime increases during the summer months compared with the winter months.

In related research, Almanie et al. [22] analyzed crime data from the United States (U.S.). cities: Denver, Colorado, Los Angeles, and California. They compared the proportion of crimes in both cities to the total number of crimes. Specific trends are found in both cities, such as Sunday having the lowest crime rate. Important derivations such as the safest and most infamous district are noted. They used a decision tree classifier and a naive Bayes classifier. According to the San Francisco Chronicle, Venturini et al. [23] used spectral analysis to discover temporal trends in crime in all of San Francisco's crime incidents from January 2003 to August 2016. They aimed to observe seasonal trends in crime and determine whether these patterns apply to all types of crime or differ by crime category. They incorporated spatial patterns and neighbourhood-level deviations from global trends. It will assist in developing temporal seasonality models based on weekly and monthly trends and features and determine the number of similar events that occurred in the community on the same day of the week or month in the previous year. Thus, the temporal analysis shows that trends vary by month and by type of crime.

## 2.2 Spatio-temporal crime forecasting

Time series prediction is a strategy for estimating future values based on previously observed values. Time series prediction is concerned with the relationships between different points in time within a single series. Interrupt time series prediction is used to identify changes in the evolution of a time series before and after an event that may affect the underlying variable. Time series prediction is a set of methods and processes that break down a series into components and explainable segments to identify patterns, estimations, and forecasts. Time series forecasting seeks to comprehend the underlying meaning of data points by using a model to predict future values based on established historical values. This study focuses on state-of-the-art time series techniques such as regression, ARIMA, and deep learning approaches.

**2.2.1 State-of-the-art time-series techniques for crime forecasting.** Time series regression is a statistical technique for forecasting future responses based on response history (autoregressive dynamics) and dynamics transferred from related predictors. According to experimental or observational evidence, time series regression may aid in understanding and forecasting complex systems. Time series regression is commonly used to model and forecast economic, social, and biological systems.

On this basis, the regression model proposed by Yadav et al. [24] was developed using data from Indian statistics, which include data on various crimes committed over the last 14 years (2001–2014), including murder, kidnapping, and robbery. Robbery and rape are both crimes. Therefore, the crime rate in various states can be forecasted for the coming years. They primarily used four data mining algorithms to analyze crime and detect crime patterns using supervised, semi-supervised, and unsupervised learning techniques such as K-mean to create multiple groups based on high and low values in criminal records.

McClendon et al. [25] exploited linear regression, additive regression, and decision stump algorithms to analyze the same collection of limited features in communities and the crime dataset from the University of California's Erbin Warehouse and current statistical data on Lamisi's crime. In general, the linear regression algorithm outperforms the three other algorithms. In addition, the linear regression algorithm finds no randomness in the test samples (without incurring too much forecasting error).

Wulff and Shaun [26] exploited ARIMA for forecasting time series data that can be used to understand or forecast the series' evolution. The process is successfully used in various fields, including economic forecasting, marketing, and industrial development. It is best suited to short-term forecasting, but forecasting requires at least 50 observations or more.

Payne et al. [27] used a crime forecasting model based on ARIMA. A correlational analysis of devastating pandemics like the Spanish flu and Covid-19 and their impact on economic growth was presented. They discovered a strong link between unemployment and criminality. They forecasted crime in Queensland, Australia, over the next six months. However, they failed to demonstrate a strong link between crime and Covid-19. Their approach applies to grid cells but generally requires significantly more historical data and is somewhat limited in incorporating additional details into a strategy.

A similar study was conducted by Yadav et al. [28] who proposed that a time series dataset can be generated by combining "big data" management techniques and generalized linear regression for statistical analysis. The ARIMA model was used to reduce the error of the forecasting model for improving the ability to identify common crime patterns among various crime sites for the selection of criminal sites. However, the described autoregressive model only fits linear data relationships. This model also ignores the nonlinear regression aspect of time series data.

**2.2.2 Deep learning techniques for crime forecasting.** The cost of machine learning methods has decreased dramatically because of the advancement of high-performance computers. They achieved great success in various fields. Notably, deep learning has yielded promising results for different classification problems, from speech initiation to visual recognition, a relatively recent advance in AI. One area of deep learning that has received little attention is crime forecasting. Several researchers observe that deep learning is also well suited to deal with the temporal and spatial elements of a problem [4, 5, 19].

Stec et al. [29] predicted crime in Chicago and Portland by using another solution based on ANN with additional external data. Crimes in Chicago are organized according to the police beats. Each beat is arbitrarily large and contains a high standard deviation of crimes. Using this high standard deviation, they divided each type of crime into ten bins of varying sizes. The regression problem was replaced by a classification problem, which is more effective and easier to measure. They also used a teaching technique known as walk-forward training. They trained various architectures, including a fully linked network, a CNN, an RNN, and a CNN + RNN architecture. The RNN + CNN combination yields the best results. However, they did not change the structures of the various networks. Finally, they investigated whether removing external data would improve or degrade the results. They discovered that public transportation and the census are also crucial for crime forecasting in their models.

Tumulak et al. [30] demonstrated a straightforward method for forecasting crime in a spatial space by combining grid thematic mapping and neural networks. Monthly, weekly, and daily data are used to train the model. The study area, Cebu City, is divided into square grids of varying sizes, and crime incidents are then recorded in each grid cell for each snapshot. This information is then fed into the neural network, which forecasts potential crime hotspots for the next time interval. According to preliminary findings, the data snapshot is separated monthly and weekly. The model is sufficiently accurate to forecast crime areas. The grid size is approximately 1000m $*$ 1000m and 750m $*$ 750m. The best model produces an F1 score of 0.95. Although the model is simple, it may serve as a starting point for future crime modelling and forecasting studies.

Wawrzyniak et al. [20] used predictive data-based modelling techniques based on data-driven ML approaches. They utilized a deep learning architecture based on ANN to achieve a high level of forecasting. A neural network architecture for crime forecasting and relevant inputs was implemented using a Gram–Schmidt calendar for network input selection. A virtual leave-one-out test was conducted to determine the optimal number of hidden neurons for optimal performance.

**2.2.3 Hybrid techniques for crime forecasting.** Advances in the hybridization of various models and algorithms have resulted in more significant and efficient ways of handling the forecasting job, resulting in increased overall effectiveness and performance. When two algorithms are combined, the results outperform both. The combination is also much faster in terms of time cost.

Delima et al. [31] proposed hybrid MGA–KNN, MGA–NB, MGA–C4.5, and MGA–RB forecasting models to find the best model for forecasting instructional results using the obtained datasets. Before forecasting, the genetic algorithm with a crossover mating scheme was integrated with the KNN, NB, C4.5, and RB algorithms. In addition, the variables were minimized using a genetic algorithm in conjunction with the proposed IBAX operator with a rank-based selection function. After ten generations, the remaining variables are used in the forecasting algorithms. For simulation purposes, forecasting accuracies of 94, 86, 89, 85, 92, 75, and 92 are obtained. However, this hybrid algorithm requires preprocessing to generate data before the forecasting process can begin.

Liu et al. [32] proposed an LSTM–STARMA hybrid model. The crime data was complicated and divided into a random seasonal pattern. The LSTM model was used for the trend and seasonal components, and the STARMA model was utilized for the randomized components after development. This solves the problem of modelling the STARMA for nonlinear or nonstationary data. The test results demonstrated the accuracy of the model in forecasting crime. They discovered that the mean and variance fluctuated more often over time. In this model, they suggested that the size of the hidden layers is 40, the number of layers is 1, the number the time steps is 2, the number of training steps is 10,000, the size of the batch is 20, and the learning rate is 0.6. The loss for the final step is 0.5042956. Their goal was to predict the number of crimes each month.

A forecasting model that combines discrete wavelet transform (DWT) and resilient back-propagation neural network (RBPNN), which is named DWT–RBPNN, was proposed by Mao et al. [33]. They forecasted crime rates in three steps. They first used sliding window sequences as input to the historical crime dataset. Then, they used DWT to divide an initial crime rate series into several subseries. In addition, crime data must be analyzed and synthesized. After that, statistical crime rate features are extracted from the subsets to reflect the wavelet coefficient distribution. Finally, RBPNN was used to predict future patterns, and the patterns and information associated with each forecasting were constructed to obtain the final forecasting series. Their proposed model is more accurate and feasible in forecasting crime rates than a single BPNN process. However, it is susceptible to the temporal alignment of the signal.

## 3 Proposed crime forecasting methodology

This section discusses the proposed methodology for crime forecasting. Moreover, the architecture of the proposed approach with the significance of different layers is highlighted along with the algorithms. Furthermore, the hybrid methodology is presented in the different experimental setups. Most of the research in this field has focused on forecasting the number of crimes, and studies indicate that forecast accuracy varies by type of crime [23, 34]. Therefore, in this study, we divided crime according to the type of crime to improve accuracy in the predictive model. We focus on five types of the most common crimes (petit larceny, harassment, burglary, criminal mischief, grand larceny).

Although the accuracy of time series predictive models has increased in recent years, they are often unreliable. Thus, this study aims to construct a predictive model that can predict with high accuracy by using the Bi-LSTM network instead of LSTM due to its significant performance on time series data [35] and by combining it with the time-series analysis technique ES. This section also describes the procedure for extracting information from the dataset used to train the proposed crime forecasting model, as shown in Fig 1.

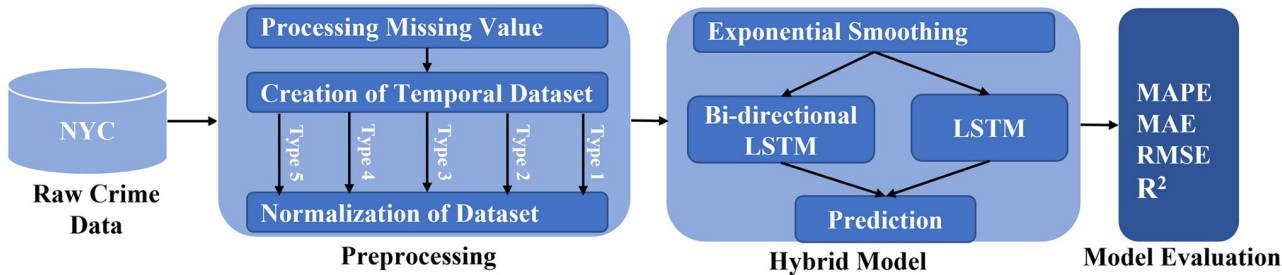

**Fig 1. Methodology for crime forecasting.**

## 3.1 Model layers

Time series is a complex problem because it depends on time and space as input parameters. Due to their repeated and sequential behavior, deep learning techniques give comparable performances in the literature. Therefore, the RNN architecture is exploited in this study to efficiently deal with sequentially dependent data. However, this method has difficulty finding or learning seasonal or level, an essential parameter of time series. Thus, ES has been used, as shown in Fig 1. The proposed methodology consists of 3 phases. First, raw NYC crime data is preprocessed to handle missing values, create temporal datasets, and normalize. Second, a hybrid model is presented to enhance prediction accuracy. Finally, four state-of-the-art evaluation measures are used to evaluate performance. The following sections discuss the workings of each layer comprehensively.

**3.1.1 Exponential smoothing layer (ES).** As previously established, the RNNs can not learn with time series components (seasonality and level). Therefore, statistical approaches outperform the RNNs, given that the time series specify the RNNs as the dataset contains seasonal data. Thus, integrating statistical technology (ES) with the neural network proposed by S.Smyl [19] in the M4 competition is advantageous for achieving the best results. As a result, a hybrid model is developed in this study.

Many statistical techniques, such as ARIMA and STL, can analyze time series. Still, ES is chosen based on M4 competition experience to obtain seasonal and level information from the transferred time series. The Holt–Winter process can be used to perform simple ES. This method is used as a pass filter to find time series components. Given that the ES coefficients are part of the neural network model, they should be included in the typical parameter fitting for the same model optimizer in our case.

Fig 2 consists of 4 stages. In the first stage, this study chooses five parallel ES and Bi-LSTM for five types of crime. ES layer is used as preprocessing layer to extract smoothing and level components. Trend component is extracted from the Bi-LSTM layer, and ensembling is performed at the individual component level for each crime type according to time series. Concatenated results are forwarded to Bi-LSTM for predicting the number of crimes. At last, time series values are denormalized, and crime trends and seasonality are forecasted.

The task ES layer is to find three main parameters as shown in the above Eqs 1 and 2 called local parameters ($\alpha, \gamma$, and $s_t$). Eqs 1 and 2 are level ($l_t$) and seasonality ($S_t$) components of the triple exponential smoothing model. Here $y_t$ is the time series value at a specific point t. Alpha and Beta are smoothing coefficients, and their values range from 0 to 1. The first Equation shows a weighted average between the seasonality and level-adjusted observations for time t-1. The second Equation presents a seasonal component for time t+m as a weighted average and

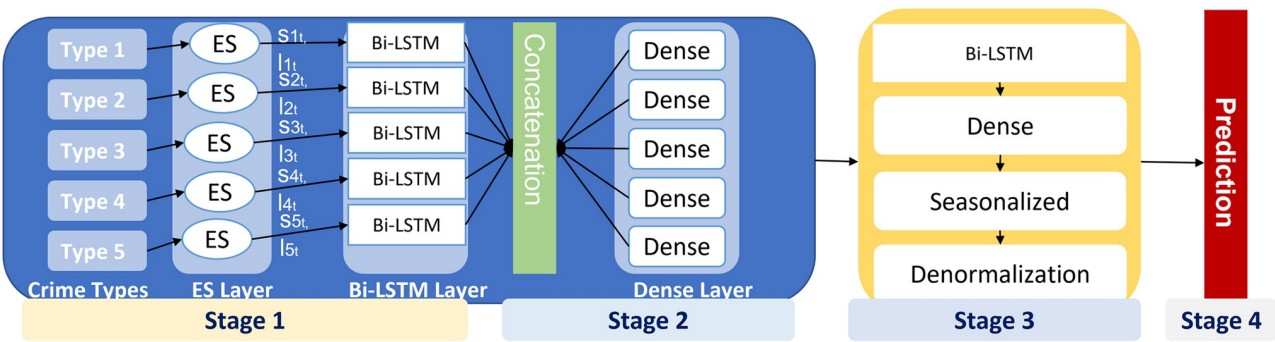

**Fig 2. Exponential smoothing and Bi-LSTM hybrid.**

predicts the seasonality component ($y_t/$, $l_t$) with the past estimate ($s_t$). These equations are derived from the Holt-Winter multiplicative model. However, we have simplified this by removing the trend component, which will cater to the Bi-LSTM model.

$$l_t = \alpha \frac{y_t}{s_t} + (1 - \alpha)l_{t-1} \tag{1}$$

$$s_{t+m} = \gamma \frac{y_t}{l_{t+1}} + (1 - \gamma)s_t \tag{2}$$

Eq 3 includes the input and output variables' current time series feature ($l_t$, $s_t$). This Eq also limits the impact of the outliers on the forecasting results. Level and seasonality components are derived from Eqs 1 and 2. Here $X_t$ is the preprocessed $t_t h$ element of the time series. $L_t$ is the last value of the level component of the input variable, and st is the $t_t h$ seasonal element.

$$X_t = \frac{y_t}{(l_t s_t)} \tag{3}$$

Local parameters can be divided into two types in ES equations:

**Local Weights**: Parameters can be learned hierarchically depending on the cost function. The network state is used to find these weights. They are called constants because they are applied to all-time series in a dataset with the same values, but they change with the cost function when a large error exists for each epoch. These weights are denoted by ($\alpha$, $\gamma$).

**Changeable Parameters**: Level and seasonality are parameters that change or learn at each step over time and are calculated using ES equations. These parameters must have initial values because they learn over time, and their default values are set based on Eq 3. Finding the default values of $s_0$ is unnecessary because it does not affect the forecasting results [36]. As shown in Eq 2, seasonality depends on the value of m. For example, in a daily forecast, the value should be equal to 7 to indicate the days of the week. As a result, the seasonal coefficient in the inner layer would be considered to be a queue of size m, and the goal of this step is to return to the same day from the previous week $s_t$ for finding $s_{(t+m)}$. This process is conducted by popping the queue from the left and then pushing $s_{(t+m)}$ from the right after calculation, as shown in Algorithm 1.

**3.1.2 Concatenate layer.** This layer is used for concatenating data from sublayer forecasting of crime types because each layer produces different data. Thus, this layer concatenates these results to be input with the final layer, and it is a non-learning layer, as shown in Fig 2. Concatenation is done by utilizing the ensembling paradigm, a well-known approach to increase performance. This layer combines time series forecasting components to produce a common response. In this study, we used the simple bagging technique for concatenating results as the simple method often performs well compared to complicated ensemble methods [37].

At the data level, forecasting produced from k models learn on $X = \{x_1, x_2, x_3, \ldots, x_k\}$ the training set is averaged. The M time series training set is split randomly into N subsets of the same size. Besides, each model learns on its own training set and forecasts each crime type's time series. The N-1 forecast of each model is averaged, and results are concatenated to produce the expected output.

**3.1.3 Seasonalized layer.** It is the final layer of the model layer; its goal is to combine the forecasting from the sub-ES layers with the final forecasting from bi-LSTM, as shown in Eq 4.

However, crime data comes from the same source but exhibits variability in seasonality patterns. Besides, starting dates according to time series are not present. Therefore, RNN alone is not sufficient to capture seasonality effectively. Consequently, we ensembled the seasonality component of all types of crime. In addition, we denormalize at the post-processing stage with the combination of ES. In contrast to [19], we denormalize values of the time series after applying normalization at preprocessing level so that small details can not be overlooked. That's why the proposed methodology outperforms as compared to state-of-the-art systems.

$$Bi - LSTM(X_t) * ((s1_t * l1_t) + (s2_t * l2_t) +$$

$$(s3_t * l3_t) + (s4_t * l4_t) + (s5_t * l5_t)) \tag{4}$$

Where 1–5 indicates the forecasting of ES for five types of crime. The inputs to this network are five, and it is also a non-learning layer. Algorithm 1 is derived from the Holt-Winter seasonality model and consists of two phases. Input to the algorithm is the Spatio-temporal time series crime data according to crime type. The primary aim is to extract level and seasonality components from the time series data. Seasonality ($S_t$) and level ($L_t$) components are extracted with weights in the first phase. Line 2-8 calculated these two parameters with alpha and gamma values. After that, data is normalized parallelly for each crime type. Line 9-28 calculated these two components' time series in different batches for each crime type and output the normalized component values of level and seasonality.

**Algorithm 1** Exponential Smoothing Layer (ES) procedure

```
1: procedure ESLayer(Input, m, BatchSize,lists)
2:   procedure build ()
3:     Let α = weight(trainable = True);
4:     Let γ = weight(trainable = True);
5:     Let SeasonalityQueue = queue[size = m, weight(Constant,
6:                       value = lists, trainable = True)];
7:     Let WeightLevel = weight(Constant, value = 0.8, trainable =
True);
8:   end procedure
9:   procedure call(Input TimeSeries)
10:     Let normalize₀ = [];
11:     Let Level_{t,0} = [];
12:     Let seasonal_{t,0} = [];
13:     l_{t-1} = WeightLevel;
14:     for i = 0 → BatchSize do
15:       y_t = TimeSeries_i;
16:       s_t = pop left(SeasonalityQueue);
17:       l_t = α (y_t/s_t) + (1 - α)l_{t-1};
18:       s_{t+m} = γ (y_t/l_t) + (1 - γ)s_t;
19:       SeasonalityQueue = push right(SeasonalityQueue, s_{t+m});
20:       normalize_i = y_t/(l_t st);
21:       Level_{t,i} = l_t;
22:       seasonal_{t,i} = s_t;
23:       WeightLevel = l_t;
24:       l_{t-1} = l_t;
25:     end for
26:     return normalize, (Level_t, seasonal_t)
27:   end procedure
28: end procedure
```

## 3.2 Hybrid model ES-BiLSTM

The hybrid model is divided into four stages:

**First stage**: forecasting the number of crimes committed for each type. It is divided into five sections, each of which forecasts the number of crimes for a specific type and includes one layer of ES, one bi-LSTM layer with six units equal to the size of the moving window, and one dense layer.

**Second stage**: Forecasting for the five types of crime in combination. It is made up of a concatenated layer of five different types of crime forecasting data from the previous stage.

**Third stage**: Forecasting the total number of crimes. It consists of a bi-LSTM layer with five units equal to the number of crime types and one dense layer.

**Fourth stage**: De-seasonalized and normalized layer, as shown in Algorithm 2.

The first stage produces five forecasts, five seasonal, and a level for each crime type. In the second stage, forecasting the five types is concatenated and transferred to the next stage. The third stage forecasts the total number of crimes. The final stage combines forecasting from the third stage with seasonality and a level for the five crimes in the first stage, as shown in Fig 3.

It is evident from Fig 3 that time-series data is forwarded to the model's first layer, consisting of three steps. First, ES smooths the time-series data to find trends and seasonality. Second, normalization is performed to remove redundant data and rescale it from 0 to 1. Finally, de-seasonalization is performed to remove short-term variations and extract underlying patterns. In the second layer, GRU is utilized to coherently extract long-term dependencies and forward them to the dense layer. The dense layer results in a temporal response along with spatial elements. Finally, de-normalization and seasonalization are performed to output future time-stamp events. The ES is used every 24 hours with an epoch of 10. Thus, the batch size of 48 times the sample will be used to increase the learning speed, as shown in Table 1.

**Algorithm 2** ES-BiLSTM hybrid detailed procedure

```
1: procedure ESLayer(Input1, Input2, Input3, Input4, Input5, m = 24,
BatchSize = 48)
2:   (NormalizedInputType1, DenormalizationCoeffType1) =ES (Input1,
m, BatchSize);
3:   LearnType1 = BiLSTM(NormalizedInputType1, hidden Unit = 6);
4:   PredictionType1 = Dense (LearnType1, hidden Unit = 1);
5:   (NormalizedInputType2, DenormalizationCoeffType2) =ES (Input2,
m, BatchSize);
```

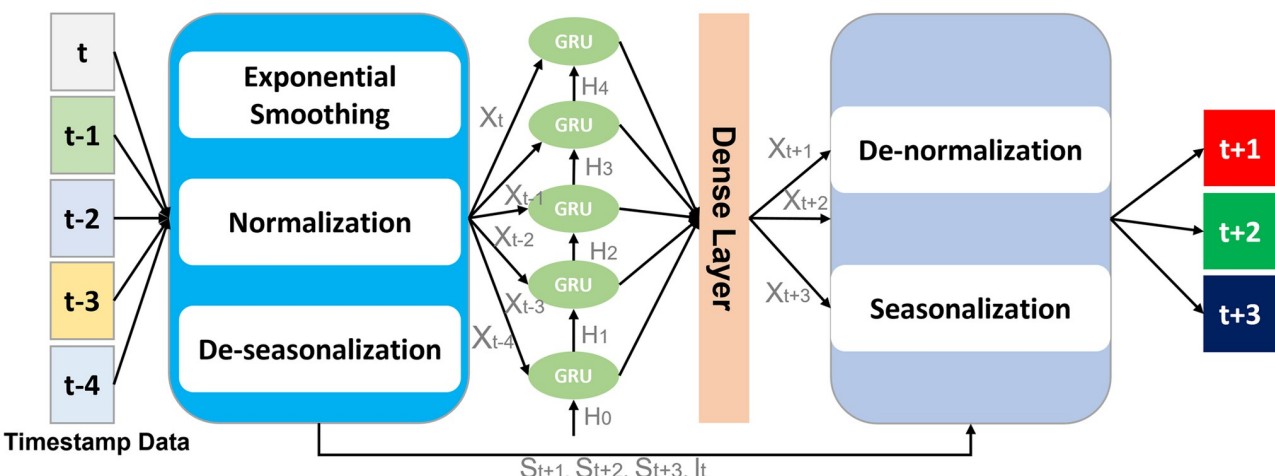

**Fig 3. Architectural details of bi-directional LSTM and ES hybrid.**

**Table 1. Hyperparameters for hourly prediction model.**

| Parameter | Properties |
|---|---|
| Seasonal | 24 |
| Batch size | 48 |
| Epoch | 10 |
| Optimizer | RMSProp |
| Lost Function | MSE |

```
 6:   LearnType2 = BiLSTM(NormalizedInputType2, hidden Unit = 6);
 7:   PredictionType2 = Dense (LearnType2, hidden Unit = 1);
 8:   (NormalizedInputType3, DenormalizationCoeffType3) =ES (Input3,
m, BatchSize);
 9:   LearnType3 = BiLSTM(NormalizedInputType3, hidden Unit = 6);
10:   PredictionType3 = Dense (LearnType3, hidden Unit = 1);
11:   (NormalizedInputType4, DenormalizationCoeffType4) =ES (Input4,
m, BatchSize);
12:   LearnType4 = BiLSTM(NormalizedInputType4, hidden Unit = 6);
13:   PredictionType4 = Dense (LearnType4, hidden Unit = 1);
14:   (NormalizedInputType5, DenormalizationCoeffType5) =ES (Input5,
m, BatchSize);
15:   LearnType5 = BiLSTM(NormalizedInputType5, hidden Unit = 6);
16:   PredictionType5 = Dense (LearnType5, hidden Unit = 1);
17:   InputFiveTypesPrediction = Concatenate (PredictionType1, Predic-
tionType2, PredictionType3, PredictionType4, Prediction Type5);
18:   AllTypesLearning = BiLSTM (InputFiveTypesPrediction, hidden
Unit = 5);
19:   AllTypesPrediction = Dense (AllTypesLearning, hidden Unit = 1);
20:   FinalPrediction = Seasonalization (AllTypesPrediction, Denorma-
lizationCoeffType1,DenormalizationCoeffType2, DenormalizationCoeff-
Type3, DenormalizationCoeffType4, DenormalizationCoeffType5);
21: end procedure
```

## 4 Experimental evaluation

This section describes the dataset used for experimental evaluation. Moreover, the necessary pre-processing steps are discussed to make the dataset reliable and consistent. Besides, the significance of the proposed LSTM and Bi-LSTM models hybrid with ES is highlighted. Furthermore, evaluation measures are explained to evaluate the proposed model. Finally, we compare and analyse the performance of the proposed model against input data sets under different experimental setups.

### 4.1 Experimental dataset

This dataset is collected from the New York City Police Department (NYPD) Complaints Dataset [38]. The dataset contains records of various police complaints from 1980 to 2020. This study uses eight years of crime data for training and testing from 2010-2017. It contains over 800,000,000 crime records (rows) with 35 attributes. However, most of these cases follow due process in filing a complaint. For this study, the most useful features of the dataset are the accident date (day, month, and year formats) and the time the event occurred (minute and hour formats). History is needed to construct the temporal base, and crime-related features are required to obtain the same pattern. This study focuses on five types due to the high number of crimes committed in these categories.

**Table 2. Depiction of NULL value in the dataset.**

| CMPL-DT | CMPL-TM | OFNS-DESC | Dtype |
|---|---|---|---|
| 433 | 36 | 0 | Int64 |

## 4.2 Pre-processing of the dataset

Pre-processing is essential to make the dataset accurate, complete, consistent, and reliable. We found several anomalies, null or missing values, consistency issues, and completeness issues in the dataset. Therefore, it is necessary to pre-process the dataset before applying the proposed model to make results reliable and accurate. Mainly, we removed Null values, made temporal data for hourly and monthly prediction, added spatiotemporal information, and normalized.

**4.2.1 Missing value.** As demonstrated in Table 2, the dataset contains null values. CMPL_DT showed the date when the crime occurred. CMPL_TM depicts the time when the crime happened. OFNS_DESC describes the crime with a specific NULL code as shown in Table 2. Last, DType is the data type of the crime with a particular code, Integer. A row with missing values cannot be deleted because it may contain critical information. Some dates and times are missing given that the data are sequential based on the date of the caller's complaint in Police Agent Service, as shown in Algorithm 3. However, the missing data could be guessed by knowing the previous and next date from the record position. If the condition is not fulfilled, then missing values could be handled by calculating the mean.

**Algorithm 3** Handling Missing Value of the Dates

```
1: procedure HandlingMissingValue(Data)
2:    for i = 1 → n do
3:      if Date_i == Null and Date_{i-1} == Date_{i+1} then
4:        Date_i = Date_{i+1};
5:      else
6:        Date_i = Date.Mean();
7:      end if
8:    end for
9: end procedure
```

**4.2.2 Creation of temporal dataset.** The goal is to create time series for capturing these data and incorporate them into the neural network's learning process to predict the number of crimes. As a result, crime incidents need to be grouped by hour of the day and day of the month. The following details are collected:

1. Each day's number of crimes per hour: Each type has crimes, and the number of crimes that occurred in a specific type per hour is determined.

2. Date: It includes a temporary series of five types of crime.

A Python program that calculates the number of records for each span of a date limit and stores it in a variable is used to obtain the number of crimes per hour associated with each record. In addition, the data would then be filtered based on the type of crime when the data do not match. The final temporal dataset now has five columns, each of which contains the number of crimes of the same type. Thus, a similar pattern of crimes is generated as shown in Algorithm 4.

**Algorithm 4** Generate Date no Crime

```
1: procedure GenerateDateNoCrime(TimeOfCrime)
2:    TimeNoCrime = Creat Time Series Range From and ((hh: mm, dd, mm,
   yyyy) → (hh: mm, dd, mm, yyyy));
3:    N = Length of TimeNoCrime;
```

```
4:    for i = 1 → n do
5:      if TimeNoCrime_i ≠ TimeOfCrime_i then
6:        InsertTimeOfCrime_i,     andNumberOfCrime = 0;
7:      end if
8:    end for
9: end procedure
```

**4.2.3 Normalization of dataset.**  Normalization is a technique that is frequently used when preparing data for ML. Normalization aims to change the values of the number columns for data mapping f use the same range without distorting differences in values times or losing information. Normalization is also required for some algorithms to perform proper data modelling. This pre-processing step is used to normalize the dataset. It provides several options for converting numerical data, such as changing all values from 0 to 1, as shown in Eq 5. Normalization can be used on single or multiple columns in the same dataset.

$$z = \frac{x_i - min(x)}{max(x) - min(x)} \tag{5}$$

## 4.3 Evaluation metrics

In this paper, four state-of-the-art forecasting KPIs have been used as evaluation measures for determining forecasting accuracy [4, 6, 20, 29]. They are used because the problem is regarded as a regression forecasting problem. In other words, this model seeks to forecast the number of crimes over time.

First, MAPE calculates the sum of individual errors divided by each period. It is the average absolute percentage error between actual ($y_t$) and predicted ($y'_t$) values, as shown in Eq 6. Moreover, it calculated the average residuals in the dataset. Second, MAE calculated the absolute error between actual ($y_t$) and predicted ($y'_t$), as shown in Eq 7. Third, RMSE is the combination of bias and variance of the prediction, and it is easy to interpret the model accuracy, as shown in Eq 8. It calculated the average of the squared difference between actual ($y_t$) and predicted ($y'_t$) values by taking the square root. It also calculated the standard deviation of residuals. Finally, $R^2$ is used to depict how well the dataset is fitted to the data. Moreover, it captures the variation between the actual and predictor response variables, as shown in Eq 9.

$$MAPE = \frac{1}{n} \sum_{t=1}^{n} \left| \frac{y_t - y'_t}{y_t} \right| \tag{6}$$

$$MAE = \frac{1}{n} \sum_{t=1}^{n} |y_t - y'_t| \tag{7}$$

$$RMSE = \sqrt{\frac{1}{n} \sum_{t=1}^{n} \left( \frac{y'_t - y_t}{n} \right)^2} \tag{8}$$

$$R^2 = \frac{SS_{RES}}{SS_{TOT}} \tag{9}$$

## 4.4 Training and testing dataset

Crime data from New York City from 2010–2017 is collected from the NYC open portal. The first four years of data (2010–2013) are used for training and testing with a ratio of 80%

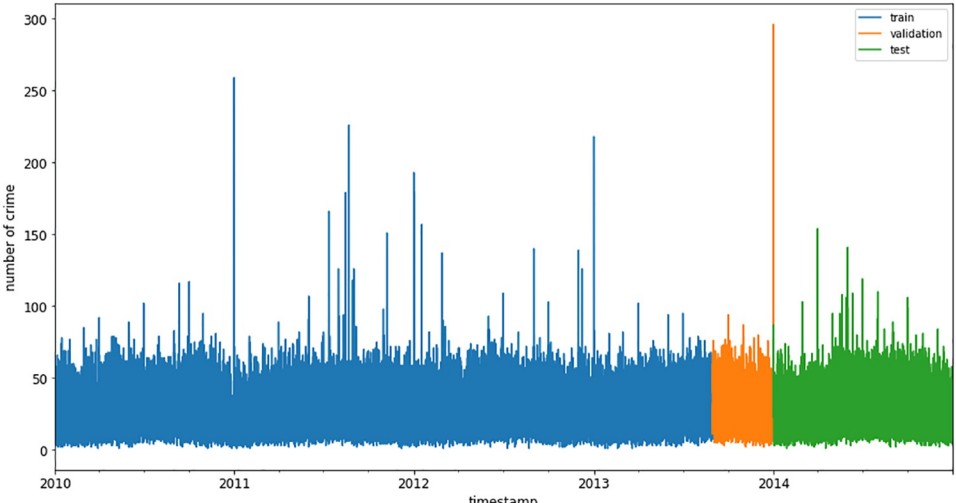

**Fig 4. Training, validation, and testing for time-series forecasting.**

training and 20% testing. In addition, the last four months' data of 2013 is used as a validation set. The first four years of data forecast the next year's (2014) crime, as shown in Fig 4. The sliding window method trains the model in different experimental setups.

**4.4.1 Moving window strategy.** A sliding window of size k created from the $x_1$ through ($x_{k-1}$) time steps is used to forecast the $x_k$ time step in the future. Each sliding window data sample has k time steps designated by ($x_1, x_2, x_3, \ldots, x_{k-1}$) that are used to forecast $x_k$ in the future. Prediction is conducted using k time steps for each type of crime, as shown in Fig 5. On the x-axis, years of NYC crime data are depicted, and on the y-axis, the number of crimes. The small circles in Fig 5 are actual data points, and the red circles show the forecast. Moreover,

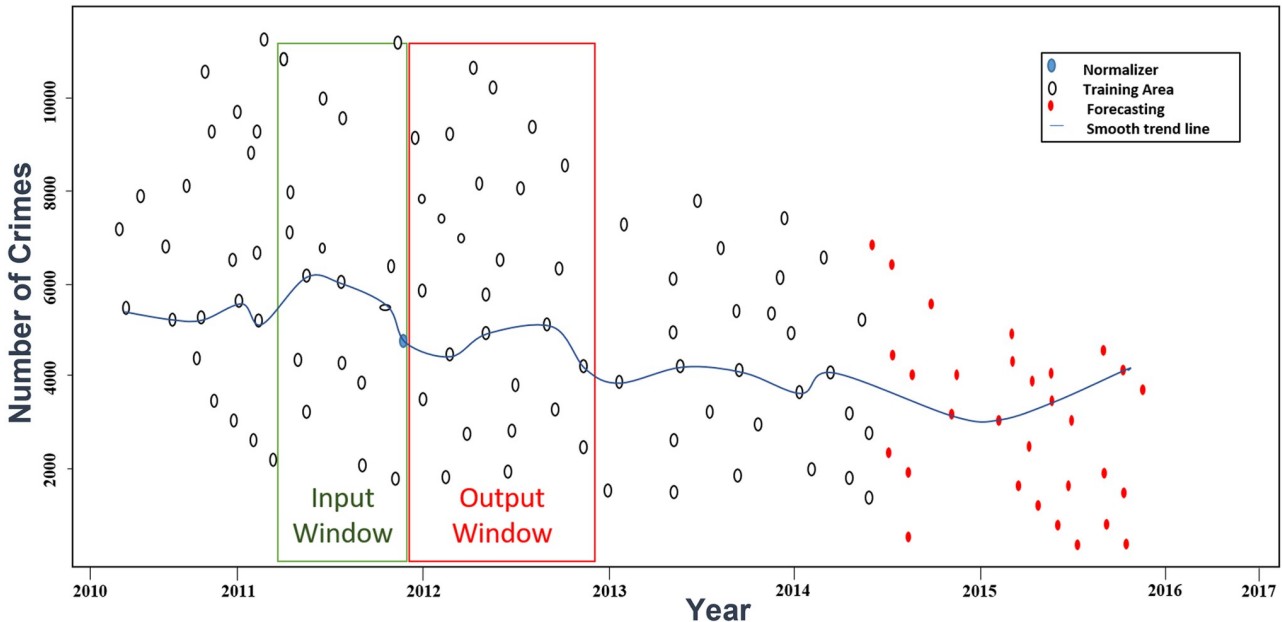

**Fig 5. Sliding window strategy for selecting the best input.**

the blue line shows the trend in the data. The value of k is set equal to 7 steps; that is, the number of data entered is six, the forecasting is at hour seven, and then the window is shifted one step forward.

Increasing the input window size and extracting more sophisticated elements, such as seasonality density and uncertainty, is a good idea; however, it adds to the computational cost. Thus, an experiment would be conducted to determine the optimal input window, which should not be larger than the seasonal size. For example, the input window in the hourly forecast ranges from 6 to 24.

## 5 Results and discussion

### 5.1 Result of selecting moving window

This experiment aims to find the best input window and determine if increasing the size improves model forecasting. The ES-BiLSTM model is applied to select an input window for each hour. BiLSTM requires input values to forecast future time series crimes. The input values of the proposed study are crime series, represented by the parameter W. The input layer (i.e., the first layer) of the BiLSTM of this study could be 6, 12, 18, and 24 in the hourly type. These values refer to the number of input parameters represented by the parameter W. For example, W6 refers to 6 input time series used in the input layer of the BiLSTM to forecast the next crime (i.e., the seventh crime). In contrast, W12 refers to 12 time-series input values to the input layer to forecast the next crime (i.e., the thirteenth crime). As shown in Table 3, an ES-BiLSTM model with a 6-input window up to a 24-input window and a six-increment amount would be used. The sample spans four years, from 2010 to 2013, and the forecast for the following year, 2014, and the input window for the remaining years would be chosen on this basis.

Table 3 shows the superiority of input window W of 6 in RMSE and MAE (13.146 and 9.837, respectively) and window W of 18 in MAPE (0.3612). However, the execution time of window sizes 6 and 18 are 00:14 and 00:30, respectively. Therefore, the window size W of 6 would be chosen because it takes less time and is better in evaluation metrics than the remaining windows.

### 5.2 Experiment of forecasting the number of crimes hourly

The results of forecasting two models (ES-LSTM and ES-BiLSTM) using two different methods will be compared in this section. The first method uses the data for the number of crimes without including the number of crime types with the model (ES-BiLSTM). The second method uses the two models with the inclusion of the number of crime types to predict the total number of crimes with the models (ES-LSTM and ES-BiLSTM). Then the results of the most promising state-of-the-art ARIMA model proposed by Catlett et al. [4] are compared and discussed in this section, as shown in Table 4, along with the results of the proposed model.

**Table 3. Results of using various moving window sizes hourly.**

| Window | MAPE | RMSE | MAE | Time |
|--------|------|------|-----|------|
| W6 | 0.3738 | 13.146 | 9.837 | 00: 14 |
| W12 | 0.3739 | 13.599 | 11.057 | 00: 18 |
| W18 | 0.3612 | 13.532 | 11.064 | 00: 30 |
| W24 | 0.3784 | 14.169 | 11.361 | 00:46 |

**Table 4. The results of forecasting the number of crimes hourly for the two models using two methods.**

| Year | Model | Type Included | MAPE | RMSE | MAE | $R^2$ |
|---|---|---|---|---|---|---|
| 2014 | ES-BiLSTM | no | 0.8542 | 21.68 | 18.366 | 0.424 |
| | ARIMA | | 6.19 | 108.34 | 88.86 | NA |
| | ES-LSTM | yes | 0.4431 | 14.316 | 11.35 | 0.4044 |
| | ES-BiLSTM | | 0.3738 | 13.146 | 9.837 | 0.6214 |
| 2015 | ES-BiLSTM | no | 0.501 | 18.295 | 15.263 | 0.363 |
| | ARIMA | | 5.42 | 97.77 | 74.54 | NA |
| | ES-LSTM | yes | 0.5529 | 17.112 | 13.752 | 0.0080 |
| | ES-BiLSTM | | 0.3891 | 13.669 | 10.896 | 0.5395 |
| 2016 | ES-BiLSTM | no | 1.1194 | 21.348 | 18.378 | 0.503 |
| | ARIMA | | 6.29 | 115.16 | 81.47 | NA |
| | ES-LSTM | yes | 0.3935 | 15.015 | 12.442 | 0.2483 |
| | ES-BiLSTM | | 0.3433 | 13.104 | 10.598 | 0.5751 |
| 2017 | ES-BiLSTM | no | 1.0692 | 17.843 | 19.318 | 0.527 |
| | ARIMA | | 7.1 | 112.05 | 79.35 | NA |
| | ES-LSTM | yes | 0.3735 | 14.141 | 11.118 | 0.2538 |
| | ES-BiLSTM | | 0.3312 | 12.529 | 9.697 | 0.6874 |

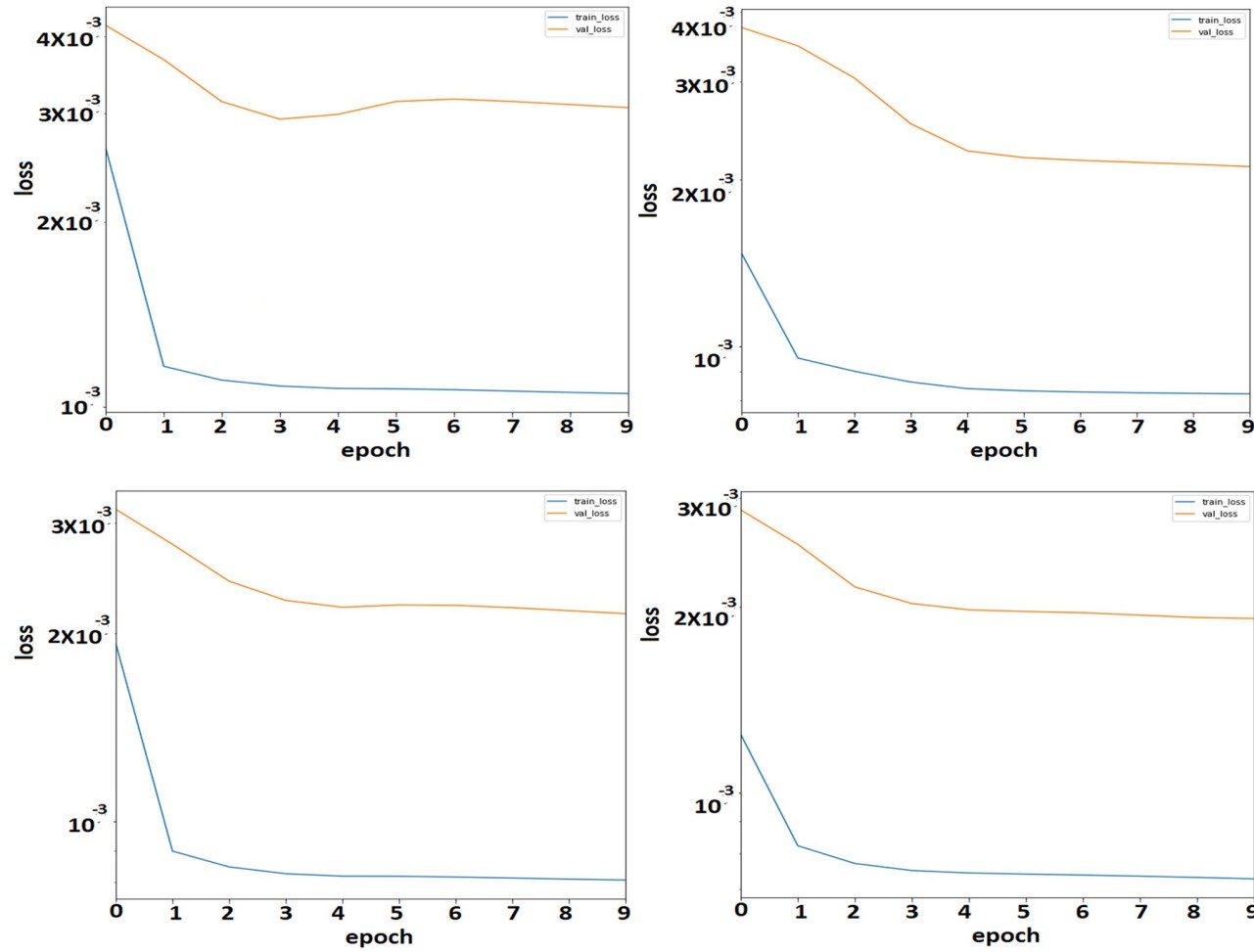

**Fig 6. Hourly loss plot between training and evaluation without including the number of crime types in ES-BiLSTM.**

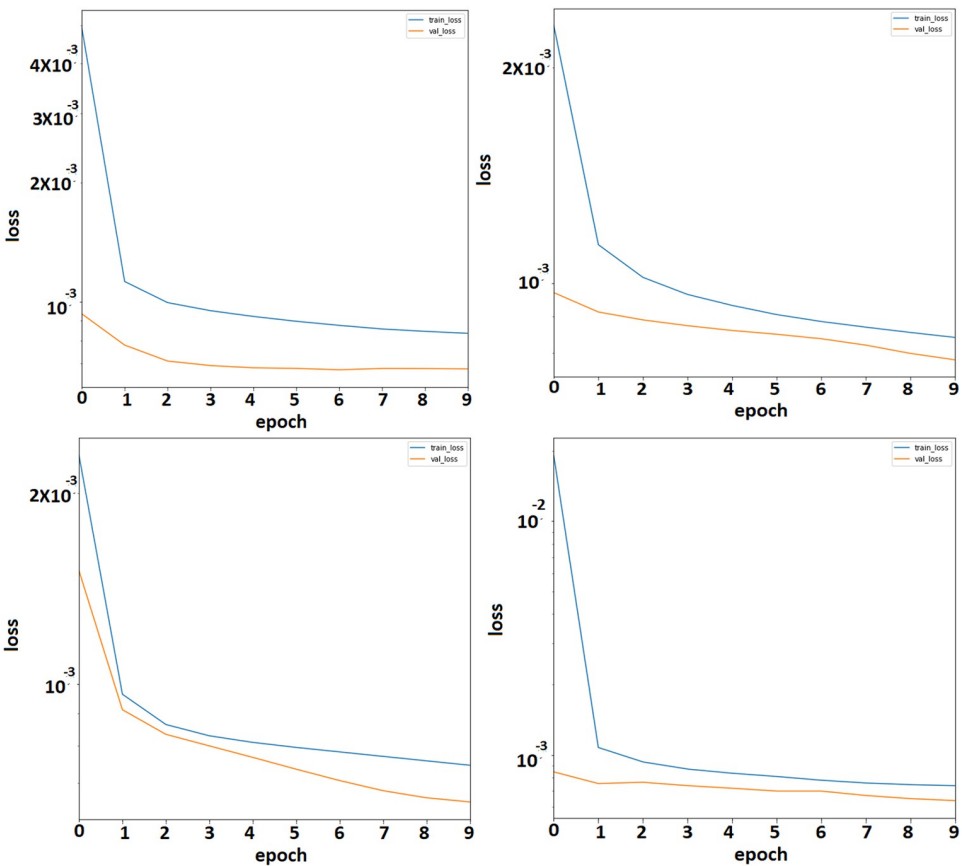

**Fig 7. Hourly loss plot between training and evaluation by including the number of crime types in ES-BiLSTM.**

Thus, as discussed in Section 5.1, the input window size would be 6. This experiment shows that the proposed method is superior to the other methods listed in Table 4. Furthermore, it displays the results of the proposed method and model in misleading fields. ES-BiLSTM, with the inclusion of the number of crime types, outperforms the other methods in accuracy measures over three years (2014–2017) because it obtains (0.3738, 0.3891, 0.3433, 0.3964) in the MAPE scale, (13.146, 13.669, 13.104, 13.77) in the RMSE scale, and (9.837, 10.896, 10.598, 10.721) in the MAE scale. Furthermore, when looking at R-squared, the proposed model ES-BiLSTM with crime types included performs comparatively well compared to other models, with values greater than 50%.

Moreover, this study also presents a training and validation plot to show the superiority of the proposed ensemble method. Figs 6 and 7 show the training and validation losses without including crime types and with including crime types. The blue line shows the training loss, and the orange line shows the validation loss. It is evident from the plots that the ES-BiLSTM method, including crime types, outperforms, as demonstrated earlier.

**5.2.1 Seasonality hourly.** The data (actual and predicted) for the same hour of the year are combined to determine the model's performance in capturing seasonality. The seasonality of the BiLSTM model is compared with and without the number of crime types included. Moreover, the LSTM model captures randomness rather than seasonality. It is evident from Fig 8 that the proposed model is relatively poor in capturing seasonality in 24 hours, and its performance appears consistent across all four years. Moreover, crime type is not included in

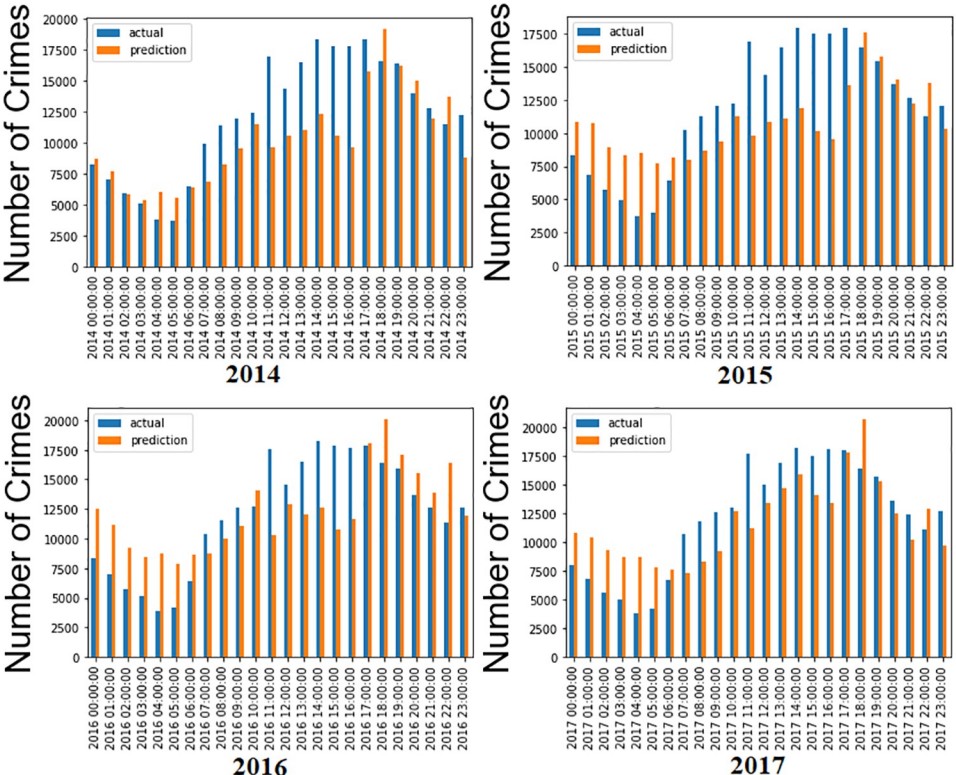

**Fig 8. Histogram of seasonal, hourly crime prediction without including crime types.** The blue color refers to the actual, while the orange color refers to the forecasting data.

this histogram for hourly crime prediction, which results in a significant difference between actual and predicted values.

The performance is much better in capturing seasonality for the four years when the number of crime types is included in the proposed method, as shown in Fig 9. In addition, 2014 and 2017 have better forecasting results than the other years. It is evident from the histogram that the actual and predicted value difference is less compared to Fig 8 when crime types are included. The proposed method demonstrated to be the best with the crime type included. Therefore, this study tests this model on daily rather than hourly data in the following section to improve the results.

## 5.3 Comparison of the proposed approach with state-of-the-art

We have chosen state-of-the-art New York City crime data [38] that has been used in several research articles. However, we now compare the results of this study under the same experimental setup with Random forest [39], ARIMA [4], SARIMA [5], and ZeroR [40]. To compare the results with state-of-the-art approaches, the same regions of New York City are used. For each technique, results are extracted by fine-tuning their hyperparameters to achieve the best results. The MAE evaluation metric is chosen in all state-of-the-art studies to compare performance. The evaluation window for all the techniques is ensured to be the same. Experimental results show the proposed approach's superiority over the state-of-the-art approaches, as shown in Fig 10.

Although this study enhanced the accuracy of crime forecasting, it has certain limitations. One disadvantage of this approach lies in its architecture and way of ensembling at various

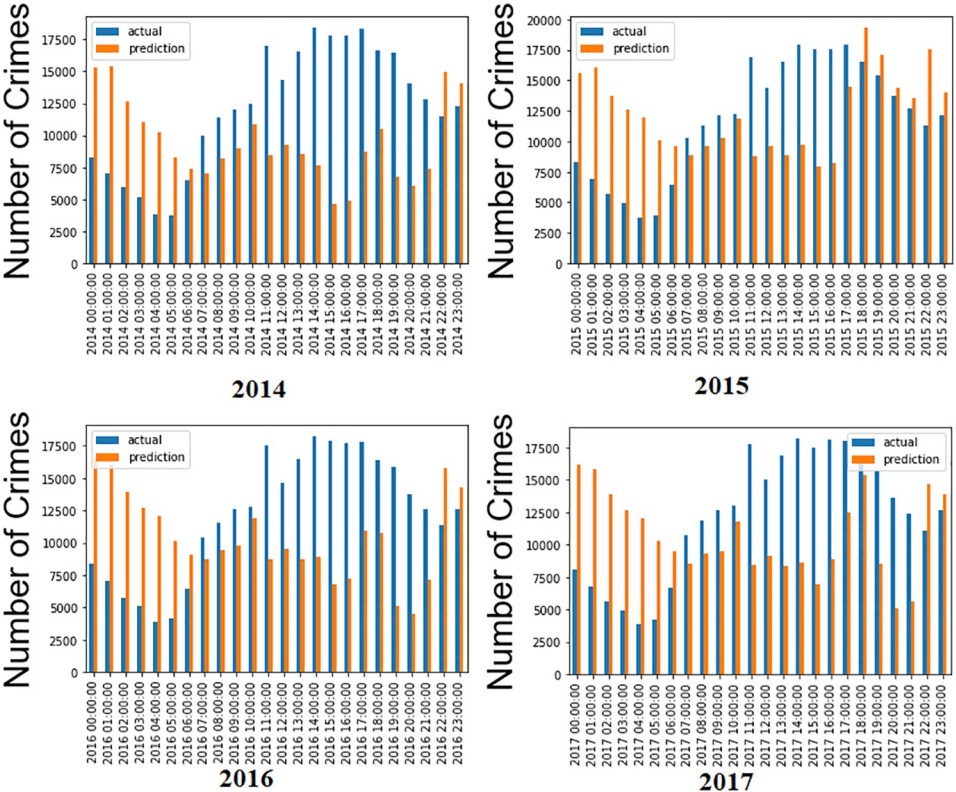

**Fig 9. Histogram of seasonal, hourly predictions with including crime types.** The blue line refers to the actual, while the orange line refers to the forecast data.

levels. Besides, fine-tuning hyperparameters is time-consuming. In the future, stochastic gradient descent may be used for optimization. Moreover, transfer learning can also be applied.

## 6 Conclusion

Crime forecasting is a crucial problem that can help law enforcement agencies to control and prevent crime. Therefore, this paper proposes a novel hybrid of Bi-LSTM and ES to forecast

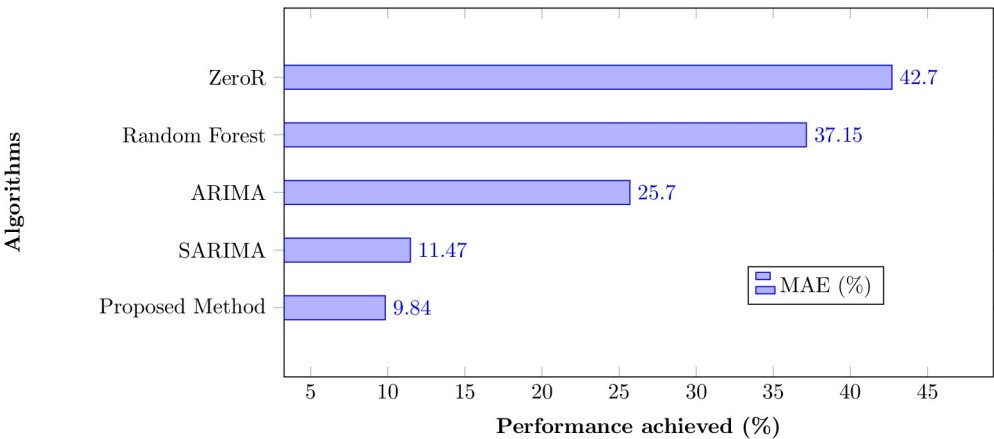

**Fig 10. Comparison of the proposed approach with state-of-the-art methods.**

crimes hourly and monthly. The proposed method is evaluated on a publicly available New York City crime dataset from 2010–2017. The first four years of crime data are used for training and testing to forecast crimes. Besides, a moving window strategy is used to select the best input parameters in terms of computational model and error measure for the proposed hybrid model. Finally, a comparative analysis is performed with state-of-the-art approaches in different experimental setups.

ES-BiLSTM, with the inclusion of crime types, outperforms the state-of-the-art by obtaining (0.3738, 0.3891, 0.3433, 0.3964) in the MAPE scale, (13.146, 13.669, 13.104, 13.77) in the RMSE scale, and (9.837, 10.896, 10.598, 10.721) in the MAE scale inaccuracy for four years (2014–2017) of data. In addition, when looking at R-squared, the ES-BiLSTM with crime types included also performed comparatively well. The proposed method could help law enforcement agencies by forecasting crime trends and patterns. Moreover, the proposed method is efficient enough for forecasting in several real-world time series applications such as weather, electricity, and finance.

We aim to exploit reinforcement learning to understand crime incident linkage in the future. Moreover, transfer learning can also enhance crime prediction accuracy by utilizing the crime domain knowledge in the relevant area. Besides, crime dataset completeness, reliability, and accuracy are vital for an efficient and reliable forecasting model.

## Author Contributions

**Conceptualization:** Umair Muneer Butt, Fadratul Hafinaz Hassan, Tieng Wei Koh.

**Data curation:** Tieng Wei Koh.

**Formal analysis:** Umair Muneer Butt, Sukumar Letchmunan, Fadratul Hafinaz Hassan, Tieng Wei Koh.

**Funding acquisition:** Sukumar Letchmunan, Fadratul Hafinaz Hassan.

**Investigation:** Umair Muneer Butt, Sukumar Letchmunan, Fadratul Hafinaz Hassan.

**Methodology:** Umair Muneer Butt.

**Project administration:** Sukumar Letchmunan.

**Resources:** Sukumar Letchmunan.

**Supervision:** Sukumar Letchmunan, Fadratul Hafinaz Hassan.

**Validation:** Umair Muneer Butt, Sukumar Letchmunan, Tieng Wei Koh.

**Visualization:** Umair Muneer Butt.

**Writing – original draft:** Umair Muneer Butt.

**Writing – review & editing:** Umair Muneer Butt.

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
