## [Decision Letter · Decision Letter 0]

10 Aug 2022

PONE-D-22-18924Hybrid of deep learning and exponential smoothing for enhancing crime forecasting accuracyPLOS ONE

Dear Dr. Butt,

Thank you for submitting your manuscript to PLOS ONE. After careful consideration, we feel that it has merit but does not fully meet PLOS ONE’s publication criteria as it currently stands. Therefore, we invite you to submit a revised version of the manuscript that addresses the points raised during the review process. Please submit your revised manuscript by Sep 24 2022 11:59PM. If you will need more time than this to complete your revisions, please reply to this message or contact the journal office at plosone@plos.org. Please include the following items when submitting your revised manuscript:A rebuttal letter that responds to each point raised by the academic editor and reviewer(s). You should upload this letter as a separate file labeled 'Response to Reviewers'.A marked-up copy of your manuscript that highlights changes made to the original version. You should upload this as a separate file labeled 'Revised Manuscript with Track Changes'.An unmarked version of your revised paper without tracked changes. You should upload this as a separate file labeled 'Manuscript'.If applicable, we recommend that you deposit your laboratory protocols in protocols.io to enhance the reproducibility of your results. Protocols.io assigns your protocol its own identifier (DOI) so that it can be cited independently in the future. For instructions see: https://journals.plos.org/plosone/s/submission-guidelines#loc-laboratory-protocols. Additionally, PLOS ONE offers an option for publishing peer-reviewed Lab Protocol articles, which describe protocols hosted on protocols.io. Read more information on sharing protocols at https://plos.org/protocols?utm_medium=editorial-email&utm_source=authorletters&utm_campaign=protocols.

We look forward to receiving your revised manuscript.

Kind regards,

Sathishkumar V E

Academic Editor

PLOS ONE

Journal Requirements:

"This work was supported by the Ministry of Higher Education Malaysia for Fundamental Research Grant Scheme (FRGS) with Project Code:FRGS/1/2020/TK03/USM/02/1, School of Computer Sciences and University Sains Malaysia (USM)."

We note that you have provided funding information that is not currently declared in your Funding Statement. However, funding information should not appear in the Acknowledgments section or other areas of your manuscript. We will only publish funding information present in the Funding Statement section of the online submission form. Please remove any funding-related text from the manuscript and let us know how you would like to update your Funding Statement. Currently, your Funding Statement reads as follows: 

"Sukumar Letchmunan: This work was supported by the Ministry of Higher Education Malaysia for Fundamental 537 Research Grant Scheme (FRGS) with Project Code:FRGS/1/2020/TK03/USM/02/1, 538

School of Computer Sciences and University Sains Malaysia (USM)."

5. Please remove your figures from within your manuscript file, leaving only the individual TIFF/EPS image files, uploaded separately.  These will be automatically included in the reviewers’ PDF.

Reviewers' comments:

Reviewer's Responses to Questions

**Comments to the Author**

1. Is the manuscript technically sound, and do the data support the conclusions?

Reviewer #3: Yes

Reviewer #4: Yes

2. Has the statistical analysis been performed appropriately and rigorously? 

Reviewer #3: Yes

Reviewer #4: Yes

3. Have the authors made all data underlying the findings in their manuscript fully available?

Reviewer #3: Yes

Reviewer #4: Yes

4. Is the manuscript presented in an intelligible fashion and written in standard English?

Reviewer #3: Yes

Reviewer #4: Yes

5. Review Comments to the Author

Reviewer #3: In this paper, the authors apply machine learning, deep learning and transfer learning models along with word weighting and embedding schemes to generate framework to classify industry specific regulations. The various combinations of techniques used by the authors are appreciable. The paper is well organized and written. The interpretation and description of the experimental results are also explained clearly. However, this manuscript have some weak points, it should be further improved before consider for publication. Some of my observations are

1. Figures 1,2,3,7 need to be drawn clearly and explanation about each figure to be incorporated in corresponding section.

2. In figure 5, x-axis legend is not mentioned and proper explanation is required.

3. In figure 9, only 8 classes are displayed? What about other classes. Justification is required.

4. Why validation is not performed? Justify

5. Why the authors specifically choose CNN and LSTM classifiers?

6. According to the results obtained, if statistical models perform better than machine, deep and transfer learning models, then what is the use of these latest techniques for analysis? Is hypertuning of these models not done properly? Justification is required.

Reviewer #4: Indicate the numerical outcome of the proposed system in the abstract section

Introduction can be more elaborate

Why rsquared value is not considered? r2 value representsd the fitness value of a model.

Future work can be aded at the end of introduction section

Figure 1 should be simplified.

Add the aplications of the proposed work.

What is the need for hybrid or ensemble models as conventional models can yield better performance?

Explain the need and the process of proposed work in detail

6. PLOS authors have the option to publish the peer review history of their article (what does this mean?). If published, this will include your full peer review and any attached files.

Reviewer #1: No

Reviewer #2: No

Reviewer #3: No

Reviewer #4: **Yes: **Usha Moorthy

---

## [Author Response · Author response to Decision Letter 0]

20 Aug 2022

We are thankful to the respected reviewers for their valuable time and suggestions to enhance the quality of this study. We addressed all the concerns of the reviewers and incorporated in the document.

---

## [Decision Letter · Decision Letter 1]

24 Aug 2022

Hybrid of deep learning and exponential smoothing for enhancing crime forecasting accuracy

PONE-D-22-18924R1

Dear Dr. Butt,

We’re pleased to inform you that your manuscript has been judged scientifically suitable for publication and will be formally accepted for publication once it meets all outstanding technical requirements.

Kind regards,

Sathishkumar V E

Academic Editor

PLOS ONE

Additional Editor Comments (optional):

Reviewers' comments:

Reviewer's Responses to Questions

**Comments to the Author**

1. If the authors have adequately addressed your comments raised in a previous round of review and you feel that this manuscript is now acceptable for publication, you may indicate that here to bypass the “Comments to the Author” section, enter your conflict of interest statement in the “Confidential to Editor” section, and submit your "Accept" recommendation.

Reviewer #2: All comments have been addressed

Reviewer #3: All comments have been addressed

Reviewer #4: (No Response)

2. Is the manuscript technically sound, and do the data support the conclusions?

Reviewer #2: Yes

Reviewer #3: Yes

Reviewer #4: (No Response)

3. Has the statistical analysis been performed appropriately and rigorously? 

Reviewer #2: Yes

Reviewer #3: Yes

Reviewer #4: (No Response)

4. Have the authors made all data underlying the findings in their manuscript fully available?

Reviewer #2: Yes

Reviewer #3: Yes

Reviewer #4: (No Response)

5. Is the manuscript presented in an intelligible fashion and written in standard English?

Reviewer #2: Yes

Reviewer #3: Yes

Reviewer #4: (No Response)

6. Review Comments to the Author

Reviewer #2: All comments, queries and explanations are addressed and the article is ready for publication. I recommend for acceptance

Reviewer #3: In this paper, the authors apply hybrid model by combining exponential smoothing and Bi-LSTM for crime forecasting. The various combinations of techniques used by the authors are appreciable. The paper is well written. The interpretation and description of the experimental results are also explained clearly.

Reviewer #4: (No Response)

7. PLOS authors have the option to publish the peer review history of their article (what does this mean?). If published, this will include your full peer review and any attached files.

Reviewer #2: No

Reviewer #3: No

Reviewer #4: **Yes: **Usha Moorthy

---

## [Editor Report · Acceptance letter]

26 Aug 2022

PONE-D-22-18924R1 

Hybrid of deep learning and exponential smoothing for enhancing crime forecasting accuracy 

Dear Dr. Butt:

I'm pleased to inform you that your manuscript has been deemed suitable for publication in PLOS ONE. Congratulations! Your manuscript is now with our production department. 

Kind regards, 

on behalf of

Dr. Sathishkumar V E 

Academic Editor

PLOS ONE